# Think Before You Place: Chain-of-Thought Video Editing for Environment-Aware Custom Subject Integration

## Abstract

Contemporary video editing methods have achieved remarkable visual fidelity for custom subject integration, yet they fundamentally lack the capability to model causally realistic interactions between inserted objects and their environments. This limitation results in physically implausible editing outcomes, violating basic physical laws. In this work, we present ThinkPlace, an end-to-end framework that addresses these challenges by leveraging Vision-Language Models (VLM) as a reasoning brain to guide physically-aware video editing without explicit physics simulation. Our approach introduces three key innovations: First, we develop a VLM-guided chain-of-thought reasoning pipeline that generates environment-aware guidance tokens while providing physically plausible editing regions for the downstream video diffusion model. Second, we introduce a Spatial Direct Preference Optimization post-training which also employs VLM for enhancing visual naturalness of editing results. Third, we leverage VLM for post-evaluation, triggering corrective refinement cycles that progressively improves integration quality. Extensive experiments demonstrate ThinkPlace achieves physically-coherent custom subject integration compared with State-of-the-art solutions. Our work represents a significant step toward bridging the gap between visual quality and physical realism in video editing applications.

## 1 Introduction

Custom subject integration is fundamental to video editing (Bai et al., 2024; Saini et al., 2024; Zhao et al., 2025). While powerful diffusion transformer (DiT) (Peebles & Xie, 2023) based methods have recently emerged, there remains substantial room for improvement. In particular, these methods directly obey user instructions for visual modifications often resulting in physically implausible outcomes, ignoring environmental constraints, or lacking proper interaction with the environment (Tu et al., 2025; Zhuang et al., 2025). For example, as shown in the left part of Figure 1, when a user requests to place a mug on a still lake surface, current models like VACE (Jiang et al., 2025) may satisfy the placement requirement by directly positioning the cup on the water. However, when users further demand physical realism, these methods fail fundamentally, as they cannot reason that ceramic mugs should sink rather than float on water. Beyond physical plausibility, extraordinary video editing requires visual naturalness: inserted objects must exhibit contextually appropriate scale and surface properties, including accurate environmental reflections and lighting responses. This motivates us to address two critical challenge: physical plausibility and visual naturalness.

Although retraining foundation models with extensive physics-oriented datasets could potentially address these limitations, such an approach would incur substantial costs in human annotation and computational resources. Moreover, as illustrated in the right part of Figure 1, requiring users to manually specify object trajectories frame-by-frame is impractical. For instance, in a ball-dropping scenario, expecting users to accurately draw the parabolic trajectory for each frame (third row) as a condition for diffusion-based editing is both tedious and technically demanding. These constraints motivate our search for a more efficient solution that leverages existing Vision-Language Models' capabilities (Bai et al., 2025), particularly Chain-of-Thought reasoning (Wei et al., 2022), to achieve environment-aware editing without expensive retraining or burdensome user input. As shown in Figure 1, our method initially interprets user instructions to conceive potential environmental

Figure 1: Thinkplace handling environment-aware custom subject integration with automatic spatial planning.

interactions within video frames, then generates semantic editing directives and physically-valid spatial regions that jointly condition downstream diffusion models (Wan et al., 2025) through an end-to-end pipeline. Crucially, we find that physical realism, unlike visual naturalness, cannot be achieved through object modifications alone: it requires adaptive environmental changes. For instance, adding supporting platforms in the lake scenario. This insight distinguishes our approach from traditional ID insertion methods, which maintain strict background invariance. **We argue that physics-driven background modifications within editing regions are essential for achieving truly realistic video editing.**

To this end, we propose ThinkPlace, a VLM-guided chain-of-thought reasoning pipeline built on diffusion transformers for custom subject integration. Our key insight is to reformulate custom subject integration from direct synthesis to a deliberate **think-then-place** paradigm. Unlike existing methods that directly execute user instructions, we introduce an intermediate reasoning stage where VLMs analyze environmental constraints and conceive physically-valid integration strategies before generation. This reasoning produces structured guidance through an Interaction Chain-of-Thought, determining not only where objects should be placed but also what environmental modifications are necessary for physical plausibility. To enhance the visual naturalness of generated results, we develop Spatial-DPO post-training with automated VLM-based preference evaluation, eliminating the need for manual annotation. We also establish iterative refinement through corrective editing, where VLM feedback progressively eliminates physical inconsistencies. We make three key contributions:

1. **Physics-aware Reasoning:** We introduce structured environmental analysis through VLM-based Chain-of-Thought reasoning, generating both semantic guidance and spatial constraints that ensure realistic object-scene interactions.

2. **Automated Spatial-DPO:** We eliminate manual preference annotation through VLM-based evaluation, enabling scalable preference optimization for enhanced physical plausibility.

3. **Automated Corrective Editing:** We employ VLM post-evaluation to trigger corrective refinement cycles that progressively enhance both insertion quality and physical plausibility.

## 2 RELATED WORK

### 2.1 VIDEO EDITING AND CUSTOM SUBJECT INTEGRATION

Video editing has witnessed rapid progress fueled by diffusion models (Ho et al., 2020; Song et al., 2021). Early efforts explored training-free (Ceylan et al., 2023; Geyer et al., 2024) or one-shot tuning (Wu et al., 2023) strategies, while subsequent methods have pursued more structured designs (Liew et al., 2023; Mou et al., 2024) to better address temporal coherence. Recently, unified and scalable frameworks have emerged: AnyV2V (Ku et al., 2024) performs first-frame editing followed by I2V propagation; VACE (Jiang et al., 2025) consolidates diverse editing tasks within a single system using Video Condition Units and context adapters; and UNIC (Ye et al., 2025)

advances task unification by representing heterogeneous inputs as tokenized sequences, enabling in-context learning without task-specific adapters. Additionally, WAN (Wan et al., 2025), a foundational diffusion transformer for text-to-video generation, has established the groundwork for diverse editing applications. While these methods demonstrate impressive versatility across multiple editing tasks, they lack specialized mechanisms for custom subject integration, particularly in modeling physically plausible object-environment interactions.

Custom subject integration, which seeks to seamlessly integrate objects from reference images into target videos, has recently attracted growing attention (Bai et al., 2024; Saini et al., 2024). VideoAnydoor (Tu et al., 2025) enhances fidelity and motion control through a pixel warper, while DreamInsert (Zhao et al., 2025) introduces a training-free paradigm for image-to-video object insertion. Moreover, Zhuang et al. (2025) substitute the conventional U-Net (Ronneberger et al., 2015) with a diffusion transformer architecture (Peebles & Xie, 2023) that leverages 3D full attention for stronger temporal modeling. Despite these advances, most existing methods overlook real-world physical constraints, often resulting in unrealistic composites. By contrast, our approach incorporates Chain-of-Thought (CoT) (Wei et al., 2022) reasoning to pre-plan insertion, leading to more natural and physically consistent results.

## 2.2 Reinforcement Learning from Human Feedback

RLHF (Reinforcement Learning from Human Feedback) (Bai et al., 2022) has become a prevalent post-training paradigm for improving large language models (Casper et al., 2023) and diffusion models through human feedback (Black et al., 2023). A notable approach under this paradigm is Direct Preference Optimization (DPO) (Rafailov et al., 2023), which directly learns from pairs of preferred and non-preferred outputs, encouraging the model to assign higher likelihoods to human-preferred results. Inspired by DPO, several methods have extended its principles to diffusion models. For instance, Diffusion-DPO (Wallace et al., 2023) introduced this framework to image generation, VideoDPO (Liu et al., 2025) adapted it to video diffusion to enhance motion fidelity and temporal coherence, and DenseDPO (Wu et al., 2025) further improved scoring by segmenting sequences for finer-grained temporal alignment. Despite these advances, current efforts have predominantly focused on video generation, with video editing remaining largely unexplored, particularly the integration of custom subjects. Moreover, existing reward formulations are limited in their ability to assess realism. To address these gaps, we propose Spatial-DPO, a variant that emphasizes the edited region and leverages vision-language models (VLMs) to provide realism-aware preference signals for optimization.

## 3 Methods

As illustrated in Figure 2, our framework consists of three key components. We first introduce a reasoning-guided architecture that leverages VLMs to perform structured environmental analysis and spatial grounding, producing both semantic and spatial guidance for integration (Section 3.1). We then present a spatial-DPO post-training approach that decomposes preference learning into local refinement and global consistency optimization, enhancing physical realism through reinforcement learning (Section 3.2). Finally, we describe our data curation pipeline that transforms real-world videos into training data through reverse engineering, enabling both connector training and DPO optimization (Section 3.3).

### 3.1 Model Architecture

#### 3.1.1 Reasoning-Guided Framework

Our framework leverages Qwen-VL2.5 (Bai et al., 2025) as Multi-Modal-Language-Model (MMLM or VLM) to perform structured reasoning over interleaved multimodal inputs comprising user instructions, reference object images, and target video sequences. This interleaved input format enables comprehensive scene understanding by establishing explicit relationships between the insertion request, object properties, and environmental constraints. As shown in Figure 2, the reasoning architecture operates through two hierarchical stages that progressively refine the integration strategy.

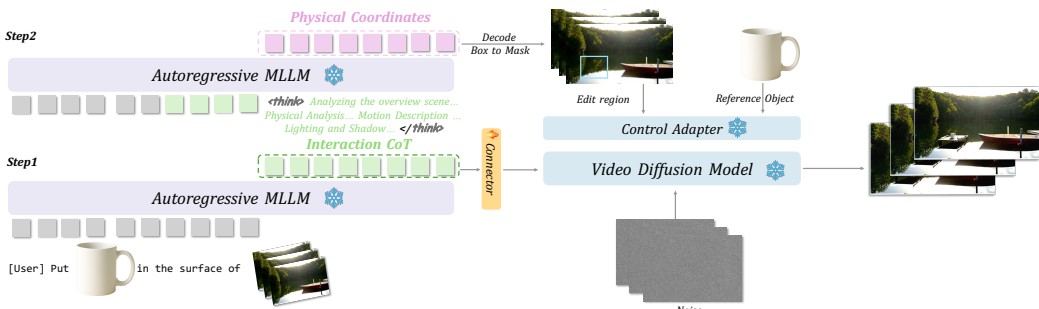

Figure 2: Inference pipeline of ThinkPlace. VLM-guided reasoning (left) which produces semantic and spatial guidance that conditions the diffusion-based generation (right).

**Global Semantic Reasoning.** In the first step, the VLM generates an Interaction Chain-of-Thought (Interaction CoT) by systematically analyzing real-world interactions within the video context. This reasoning process examines object-environment relationships to establish physically-grounded integration strategies through four key aspects: (i) scene overview for comprehensive environmental understanding, (ii) physical analysis for evaluating integration constraints and inferring auxiliary structures when needed, (iii) motion description for temporal dynamics, and (iv) lighting and shadow analysis for photometric consistency. These analyses synthesize into structured token representations encoding both interaction protocols and rich contextual information, providing the semantic foundation for downstream generation. The detailed reasoning process is provided in Appendix A.1.

**Local Spatial Grounding.** Following semantic reasoning, the second step grounds abstract interaction strategies into concrete spatial locations. The VLM takes the generated Interaction CoT as additional context alongside the original video inputs, enabling spatially-aware decision making. This two-stage design is crucial: the first stage determines *what* needs to happen (e.g., "add floating platform for mug"), while the second stage determines *where* it should occur within the frame. The VLM outputs precise bounding boxes $[x_1, y_1, x_2, y_2]$ that specify not only the target object's placement but also regions that may require environmental modifications. These coordinates are converted into binary masks that provide pixel-level guidance for the subsequent generation process.

### 3.1.2 ENVIRONMENT-AWARE CUSTOM SUBJECT INTEGRATION

Our generative pipeline builds upon the VACE (Jiang et al., 2025) framework, extending Wan2.1 (Wan et al., 2025) with enhanced multi-modal conditioning for environment-aware video editing. As illustrated in Figure 2, we integrate the reasoning outputs through two complementary conditioning pathways that work in tandem to achieve physically plausible integration with an end-to-end manner.

**Semantic Conditioning Pathway.** This pathway translates high-level reasoning into generation guidance. We design a lightweight connector module that bridges the representation gap between the VLM's reasoning space and the diffusion model's conditioning space. Specifically, the connector learns to project environment-aware Interaction CoT tokens into the T5 embedding space used by Wan2.1. During training, the connector is optimized to preserve the semantic richness of reasoning outputs while producing effective conditioning signals for the diffusion model. This allows the generated video to faithfully reflect the VLM's physical understanding, such as adding support structures or adjusting object dynamics based on environmental constraints.

**Spatial Conditioning Pathway.** While semantic conditioning provides the "what" and "how," spatial conditioning specifies the "where." The bounding boxes from spatial grounding are rasterized into binary masks that precisely delineate editing regions. These masks serve dual purposes: (i) constraining modifications to relevant areas, preserving the rest of the scene, and (ii) indicating regions where auxiliary structures or environmental changes should appear. The masks directly interface with VACE's spatial control mechanisms, ensuring that reasoning-determined modifications occur exactly where intended. This spatial precision is essential for maintaining scene coherence while enabling localized physics-aware edits.

Figure 3: Pipeline of Spatial-DPO Post-training, where global and local optimization are combined.

## 3.2 SPATIAL-DPO POST-TRAINING

We employ reinforcement learning to enhance visual naturalness in video diffusion models, specifically targeting physically plausible integration of custom subjects. Our approach introduces a Spatial-DPO post-training framework that decomposes the preference learning objective into two complementary components: local refinement and global consistency optimization. Following the methodology of Diffusion-DPO (Wallace et al., 2023), we reformulate the preference learning objective to address the unique challenges of custom subject integration in video generation. Given identical inputs comprising an object image, source video, and interaction prompt, we generate multiple candidate outputs through stochastic sampling with varying random seeds. These outputs form preference pairs for our optimization framework.

**Local Refinement DPO.** Operating across the entire video sequence, this component focuses on fine-grained optimization within edit regions defined by bounding boxes from our reasoning pipeline. By constraining DPO optimization to these localized interaction zones throughout all frames, it specifically enhances critical details where physical accuracy matters most, ensuring physically plausible object-environment interactions and natural contact dynamics at insertion boundaries. We denote this localized loss as $\mathcal{L}_{DPO}^{local}$.

**Global Consistency DPO.** Also operating across the entire video sequence, this component focuses on scene-wide optimization across the entire frame. It ensures inserted objects remain consistent in appearance throughout the video, preserves background stability in non-edited areas, and maintains overall lighting and color harmony, guaranteeing that local modifications integrate seamlessly with the full scene across all temporal frames. We denote this global loss as $\mathcal{L}_{DPO}^{global}$.

Following Diffusion-DPO (Wallace et al., 2023), the standard DPO loss is defined as:

$$\mathcal{L}_{DPO} = -\mathbb{E}_{(v_w, v_l)} \left[ \log \sigma \left( \beta \Delta_{\theta, ref} \right) \right] \qquad (1)$$

where $\Delta_{\theta, ref} = (L_\theta^l - L_\theta^w) - (L_{ref}^l - L_{ref}^w)$ with $L = \|\epsilon_\theta(v^t, t) - \epsilon\|^2$ being the denoising loss, and $\beta$ controls the preference strength.

We extend this to spatial-aware optimization by combining both components:

$$\mathcal{L}_{total} = \lambda_{global} \cdot \mathcal{L}_{DPO}^{global} + \lambda_{local} \cdot \mathcal{L}_{DPO}^{local} \qquad (2)$$

where the hyperparameters $\lambda_{global}$ and $\lambda_{local}$ balance between maintaining scene-wide coherence and achieving fine-grained interaction fidelity. This dual optimization strategy ensures comprehensive quality improvement: global consistency preserves overall video coherence and temporal stability, while local refinement precisely enhances object-environment interactions at critical contact points.

**VLM-Guided Preference Ranking.** The absence of reliable metrics for quantifying physical realism in video editing motivates our use of VLM reasoning for preference assessment. We again leverage Qwen-VL 2.5 to evaluate generated samples across multiple dimensions of physical

plausibility, including gravitational consistency, surface contact naturalness, illumination coherence, and motion trajectory validity. To address the inherent stochasticity in VLM-based evaluation, we implement a consensus-based ranking protocol. For each set of generated candidates, we conduct two independent ranking sessions with randomly permuted sample orderings. Preference pairs are accepted only when both sessions produce consistent rankings, effectively reducing evaluation noise and improving alignment with human judgments. This dual-ranking mechanism significantly reduces false positives in preference identification while maintaining computational efficiency compared to human annotation. Through this spatially-aware DPO framework guided by VLM reasoning, we achieve substantial improvements in physical realism and environmental coherence of inserted objects, as demonstrated in our experimental evaluations.

## 3.3 DATA CURATION

To train our connector module and enable DPO post-training, we construct a custom subject integration dataset. While synthetic trajectory generation using VLMs on background videos presents an intuitive approach, it suffers from two critical limitations: prohibitive computational costs for large-scale generation and the absence of ground truth for validation. We therefore adopt a reverse-engineering approach using real-world videos. Our data collection encompasses two complementary categories: (i) human-object interaction videos (10,000 samples) capturing natural manipulation and handling behaviors, and (ii) physics-demonstration videos (4,000 samples) showcasing fundamental physical phenomena including collisions, combustion, and gravitational dynamics. These raw videos undergo systematic processing through our data curation pipeline that performs VLM-based subject identification, DINO-SAM cascade for precise localization and segmentation, and Bagel-based completion to address natural occlusions. This approach transforms existing videos into paired training data consisting of reference objects and corresponding videos with subject regions masked, while also introducing beneficial appearance diversity to improve model robustness. The detailed pipeline architecture and processing examples are provided in Appendix A.2.

## 4 EXPERIMENTS

### 4.1 IMPLEMENTATION DETAILS

ThinkPlace is built upon QwenVL2.5-7B for vision-language reasoning and VACE (based on Wan2.1-1.3B) for video generation. The connector module, consisting of a two-layer MLP, trained for 500K iterations with a batch size of 16, while keeping other components frozen. For DPO post-training, we perform full fine-tuning of VACE for 10K iterations with a batch size of 8. $\beta$ is setting to 100, and $\lambda_{global}$, $\lambda_{local}$ are setting to 0.5. Notably, ThinkPlace supports flexible user interaction: users can directly specify editing regions, bypassing the automatic region generation in Step 2. When provided with user-defined masks, ThinkPlace focuses exclusively on maximizing insertion realism within the specified constraints. This flexibility is leveraged during training, where we use pre-masked videos from our dataset to eliminate Step 2 computation, improve training efficiency.

### 4.2 COMPARISONS

#### 4.2.1 QUALITATIVE COMPARISONS.

We compare ThinkPlace with state-of-the-art video editing methods UNIC (Ye et al., 2025) and VACE (Jiang et al., 2025). Since UNIC is not open-source, we utilize their publicly available demonstrations. For intuitive comparison, we initially adopt UNIC's editing regions across all methods to eliminate regional selection bias. Subsequently, we conduct experiments using our VLM-guided editing regions to demonstrate the advantages of reasoning-based spatial guidance. For VACE, we use GPT-4o generated video captions as input prompt for T5. We select test scenarios that span varying complexity levels: from simple object placement to challenging physics-based interactions involving combustion and fluid dynamics. Figure 4 demonstrats ThinkPlace's superior performance in both standard insertion tasks and complex physical scenarios.

**Top Row** Using UNIC's editing regions, ThinkPlace achieves high-fidelity insertion while enabling natural environmental interactions absent in baselines. In the donut example (red arrows), our method generates physically plausible tilting responses to approaching waves, while UNIC produces static

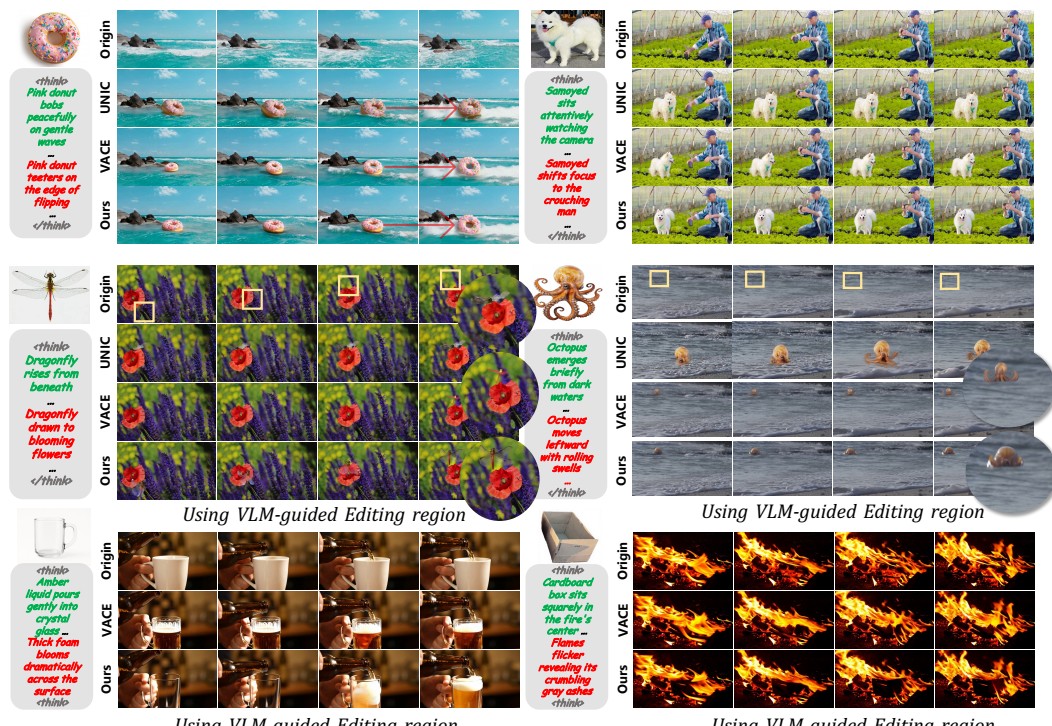

Figure 4: Qualitative Comparisons. The top tier shows comparisons using UNIC's demo editing regions. The middle and bottom tiers demonstrate methods using our VLM-guided editing regions, including VACE and our ThinkPlace (UNIC use it's own edit region for comparsions). Bottom tier: ThinkPlace demonstrates physically realistic insertions, with the left example additionally showcasing object replacement capability. The comparison illustrates ThinkPlace achieves physically plausible object-environment interactions through its reasoning-based approach.

placement. The Samoyed example further demonstrates this capability: our inserted dog naturally turns toward the person, exhibiting environmental awareness that UNIC lacks. **Middle Row** With our VLM-guided editing regions, environment-aware reasoning reaches its full potential. The dragonfly dynamically responds to the flower's presence with natural attraction behavior. In the octopus scene, our method captures ocean current dynamics, producing fluid tentacle movements that far exceed UNIC's static insertion in both realism and visual quality. **Bottom Row** The beer pouring demonstrates our replacement capability with accurate physics: liquid level decreases realistically and foam formation follows fluid dynamics, while VACE maintains static liquid volume. The combustion example highlights our state-aware insertion: ThinkPlace renders the box in its burned state with appropriate charring and deformation based on environmental context, whereas VACE fails to adapt the object's physical state, inserting the original intact box despite the fire context. These results validate ThinkPlace's extraordinary capability in achieving both visual fidelity and physical plausibility in complex real-world scenarios.

### 4.2.2 QUANTITATIVE COMPARISONS.

We conduct comprehensive quantitative evaluation on 200 test videos comparing ThinkPlace with the state-of-the-art VACE (Jiang et al., 2025) (UNIC not open-source) and our ablated variants, as shown in Table 1. We evaluate across three dimensions: Identity Preservation (CLIP-I (Radford et al., 2021) and DINO-I (Caron et al., 2021)), Video Quality (Huang et al., 2024) (temporal smoothness and aesthetics). Physical Realism measured using the state-of-the-art VideoPhY benchmark (Bansal et al., 2025), which quantitatively assesses Physical Commonsense (PC) and Physical Rules (PR) adherence, providing objective and reproducible evaluation of physical plausibility in generated videos. ThinkPlace consistently achieves the best performance across all metrics. While improvements in

| Method | Identity | | Video Quality | | Physics | |
|---|---|---|---|---|---|---|
| | CLIP-I ↑ | DINO-I ↑ | Smooth. ↑ | Aesth. ↑ | PC ↑ | PR ↑ |
| VACE (Jiang et al., 2025) | 0.7723 | 0.4454 | 0.9810 | 0.5281 | 4.035 | 0.802 |
| ThinkPlace-CoT | 0.7718 | 0.4539 | 0.9911 | 0.5297 | 4.041 | 0.834 |
| ThinkPlace-DPO | 0.7765 | 0.4583 | 0.9898 | 0.5362 | 4.070 | 0.832 |
| **ThinkPlace** | **0.7778** | **0.4681** | **0.9925** | **0.5374** | **4.103** | **0.852** |

Table 1: Quantitative comparisons among 200 test pairs. VACE using GPT-4o's caption and keep same edit region with ThinkPlace. PC: Physical Commonsense, PR: Physical Rule.

identity preservation and video quality are notable, the most significant gains appear in physical realism metrics. Specifically, Physical Commonsense (PC) and Physical Rules (PR) scores show substantial improvements over VACE, with PC exhibiting the largest relative gain among all metrics. These results quantitatively confirm that our reasoning-guided approach successfully addresses the physical plausibility challenges that plague existing video editing methods.

### 4.3 ABLATION STUDY

We investigate the individual contributions of interaction CoT reasoning and Spatial-DPO through systematic ablation. These experiments confirm that CoT and Spatial DPO provide complementary benefits: CoT enhance physical plausibility while Spatial DPO refines visual naturalness.

**Quantitative Analysis**. To validate the contribution of each component in ThinkPlace, we conduct ablation studies by removing the Chain-of-Thought reasoning (ThinkPlace-CoT) and Direct Preference Optimization (ThinkPlace-DPO) modules separately. As shown in Table 1, removing either component leads to performance degradation across all metrics. The complete model achieves the best performance, validating the synergy between interaction CoT and Spatial-DPO.

**Qualitative Analysis.** To illustrate the importance of each component, we present a challenging real-world scenario: inserting a test tube into a beaker containing water, as shown in Figure 5. This task requires precise physical understanding and environment-aware reasoning. **Left: CoT and DPO Ablation.** The baseline VACE (without CoT or DPO) produces physically implausible results with severe artifacts: the tube unnaturally penetrates the water surface and exhibits temporal flickering. Adding CoT alone reduces water artifacts but retains tube artifact issues and unrealistic water level changes. Incorporating only DPO significantly reduces visual artifacts, yet the refracted background through the liquid remains blurry and unconvincing. The complete ThinkPlace achieves physically accurate insertion with proper water displacement, correct refraction, and temporal stability. **Right: Spatial-DPO Component Analysis.** As demonstrated in the right panel of Figure 5, we analyze the

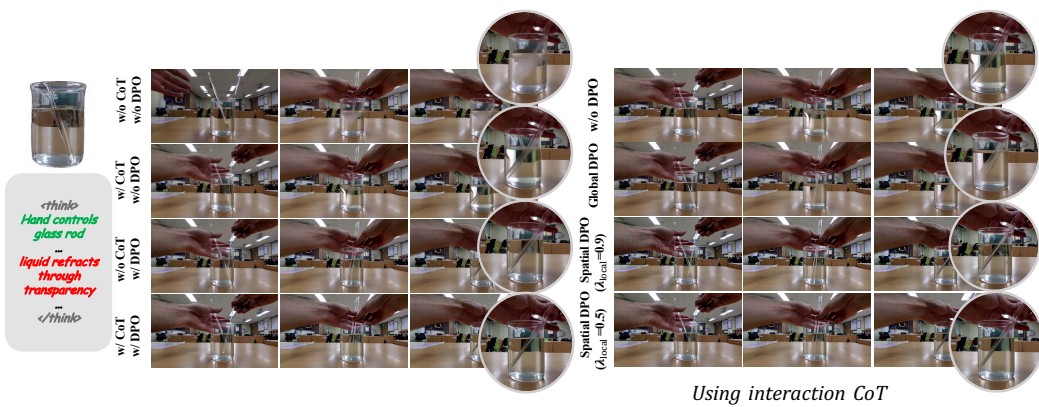

Figure 5: CoT and Spatial DPO demonstrate synergistic improvements in visual naturalness.

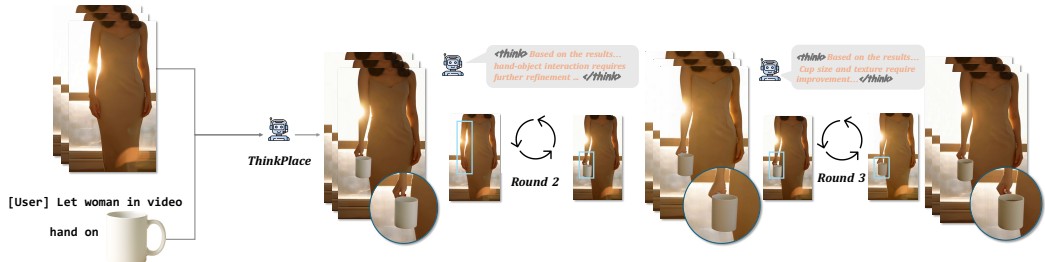

Figure 6: Corrective editing is an iterative refinement mechanism that progressively improves editing quality based on the VLM post-evaluation.

contribution of local versus global DPO optimization (all variants include CoT). Using only CoT produces basic physical plausibility but lacks fine-grained interaction details. Adding global DPO alone improves overall consistency but water-tube interaction boundaries remain blurry and imprecise. When both local and global DPO are applied with imbalanced weights ($\lambda_{local} = 0.9, \lambda_{global} = 0.1$), local details improve but insufficient global optimization causes background flickering. The full model with balanced weights ($\lambda_{local} = 0.5, \lambda_{global} = 0.5$) achieves optimal results: sharp interaction boundaries, stable backgrounds, and physically coherent water dynamics.

### 4.4 APPLICATION: ITERATIVE REFINEMENT THROUGH VLM POST-EVALUATION

We employ VLM post-evaluation to trigger corrective refinement cycles that progressively enhance both insertion quality and physical plausibility. As illustrated in Figure 6, our post-evaluation mechanism operates through systematic diagnostic-refinement loops. After each generation pass, the VLM performs comprehensive post-evaluation to identify physical violations and visual artifacts. In the cup insertion example, the initial generation exhibits deficiencies in photorealism and hand-object dynamics. The VLM's diagnostic evaluation quantifies these failure modes and automatically triggers corrective refinement, reformulating both the interaction chain-of-thought and spatial guidance. The post-evaluation continues iteratively: after the second generation improves interaction coherence, the VLM detects remaining issues in scale consistency and lighting. Only when the VLM's evaluation confirms physical plausibility and visual coherence does the refinement cycle terminate, typically achieving convergence within 2-3 iterations. This VLM-driven post-evaluation paradigm validates that complex editing tasks benefit from iterative assessment and targeted correction rather than single-pass generation. The detailed diagnostic-refinement protocol are described in Appendix A.3.

## 5 CONCLUSION

We presented ThinkPlace, a novel framework that reformulates custom subject integration from direct synthesis to a deliberate think-then-place paradigm. By introducing environment-aware VLM-guided reasoning before generation, combined with Spatial-DPO post-training and iterative corrective editing, our approach achieves both physical plausibility and visual naturalness in video editing. This environment-aware paradigm eliminates the need for expensive model retraining or manual trajectory specification, instead leveraging VLMs' understanding of physical laws to guide generation. Extensive experiments confirm that ThinkPlace outperforms state-of-the-art methods, particularly in scenarios requiring complex physical reasoning and adaptive environmental modifications.

**Limitations and Future Work.** While ThinkPlace achieves significant improvements in physical plausibility, several limitations remain. The iterative refinement process increases inference time, typically requiring 2-3 generation cycles for complex scenarios. The current framework relies on VLM reasoning quality, which may occasionally produce suboptimal guidance for highly unusual or abstract editing requests. Future work could explore several promising directions: (1) extending the framework to handle multiple interacting objects simultaneously; (2) developing more efficient architectures that reduce the memory consumption.

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

# A  APPENDIX

## A.1  REASONING PROCESS OF GLOBAL SEMANTIC ANALYSIS

The Global Semantic Reasoning component follows a structured reasoning process:

1. *Scene Overview*: The VLM-based agent first analyzes the video frames, identifying environmental elements (surfaces, boundaries, existing objects) and their spatial relationships. Concurrently, it examines the reference object's properties (material, weight, physical state) and interprets user requirements to establish a comprehensive understanding of both the scene context and integration objectives.

2. *Physical Constraint Evaluation*: Based on scene understanding, the agent evaluates physical constraints including surface properties, gravitational requirements, and contact mechanics.

3. *Scene Modification (optional)* : When direct placement violates physical laws, the agent infers necessary auxiliary structures or environmental modifications to enable physically plausible integration.

3. *Motion Dynamics Analysis*: The agent tracks camera movement and environmental dynamics to plan coherent object trajectories, ensuring temporal consistency and natural motion patterns that respond appropriately to environmental forces.

4. *Lighting and Shadow Assessment*: The agent analyzes illumination conditions, shadow patterns, and reflection properties to ensure photometric consistency of the inserted object within the scene context.

5. *Interaction CoT Generation*: Integrating outputs from the preceding reasoning steps, the agent formulates structured token representations that encode both interaction protocols and rich contextual information.

This process generates an Interaction Chain-of-Thought that serves as the semantic foundation for the subsequent generation process.

## A.2  DATA CURATION

Our data curation pipeline operates through three sequential stages as illustrated in Figure 7:

1. *VLM-based Subject Identification*: We leverage VLM capabilities to identify and extract the primary subject from each video, generating both semantic labels and reference images. For instance, when processing a kitchen scene video, the VLM correctly identifies "pineapple" as the target object and extracts its visual representation.

2. *DINO-SAM Cascade Segmentation*: We employ DINO for fine-grained localization followed by SAM for precise segmentation. DINO provides accurate bounding boxes around the identified subject, which serve as prompts for SAM to generate pixel-accurate masks. This process extracts the subject from its original context, yielding training pairs consisting of reference objects and corresponding videos with subject regions masked.

3. *Bagel-based Completion and Augmentation*: A critical challenge arises from natural occlusions in real videos, resulting in partially visible reference objects. We employ Bagel to intelligently complete occluded regions, generating multiple plausible variations of each reference object. This augmentation strategy not only resolves the incompleteness issue but also introduces beneficial appearance diversity.

## A.3  VLM DIAGNOSTIC-REFINEMENT CYCLE

The corrective editing mechanism operates as a closed-loop system where VLM diagnostics drive iterative improvements:

1. *Output Analysis*: The VLM examines the generated video across multiple dimensions including physical consistency, visual coherence, lighting integration, and interaction dynamics. This analysis produces a structured diagnostic report identifying specific failure modes.

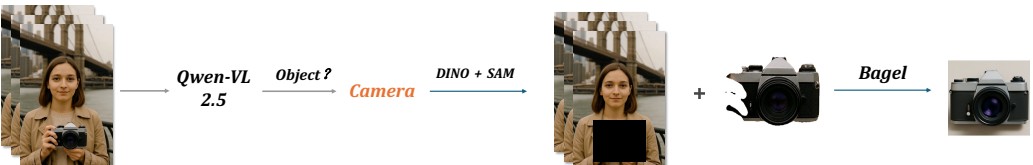

Figure 7: Data curation of ThinkPlace.

2. *Failure Mode Identification*: Based on the diagnostic analysis, the VLM categorizes issues into primary violations (e.g., impossible physics, severe scale errors) and secondary refinements (e.g., shadow softness, reflection intensity). This prioritization ensures critical issues are addressed first.

3. *Reasoning Reformulation*: The VLM updates the Interaction Chain-of-Thought to address identified issues. This involves modifying physical constraints, adjusting spatial relationships, or adding auxiliary elements to resolve violations.

4. *Spatial Region Adjustment*: Corresponding to the reasoning updates, the VLM refines bounding boxes and editing masks to better capture areas

## A.4 LLM USAGE

We used Claude for grammar checking and language polishing of the manuscript. All technical content, experimental design, and scientific insights are original work by the authors. The LLM was not involved in research ideation, experimental execution.

