# OpenReview forum: "Think Before You Place: Chain-of-Thought Video Editing for Environment-Aware Custom Subject Integration"
_ICLR.cc/2026/Conference — ICLR 2026 Conference Withdrawn Submission_

### Official Review · Reviewer_4EaR · 2025-10-19

**Soundness:** 3
**Presentation:** 3
**Contribution:** 3
**Rating:** 4
**Confidence:** 3

**Summary:**

This paper introduces ThinkPlace, a framework for environment-aware custom subject integration in video editing. The main contribution is the proposal of a "think-then-place" paradigm, where a Vision-Language Model (VLM) is used to reason about physical plausibility and environmental constraints *before* guiding a generative model.

However, the paper suffers from several significant weaknesses. First, the methodological novelty is limited; the framework is heavily based on the existing VACE model, with the primary additions being the VLM reasoning module and a new dataset created for this task. More critically, the evaluation is a significant weakness. For this newly proposed task, the paper fails to introduce a proper evaluation benchmark. The few quantitative results that are provided show only marginal improvements over the baseline, which calls the effectiveness of the complex proposed framework into question. While the paradigm itself is an interesting idea and the paper is well-written, the limited methodological contribution and weak evaluation make the paper's value doubted.

**Strengths:**

1.  The primary strength of the paper is the introduction of the "think-then-place" paradigm. This is a novel and interesting conceptual approach for addressing the complex task of environment-aware custom subject integration in videos. By explicitly adding a VLM-driven reasoning step before the generation, the framework attempts to solve the critical problem of physical plausibility, which current methods often fail to address.

2.  The paper is generally well-written and clearly structured. The methodology and the "think-then-place" concept are explained in an organized and relatively easy-to-follow manner.

**Weaknesses:**

1.  The methodological novelty of the paper is limited. The proposed framework is heavily built upon the existing VACE model, introducing a VLM-based reasoning module and a new dataset. While the "think-then-place" paradigm is a novel task formulation, the underlying generative method itself is not a new contribution.

2.  The evaluation is a significant weakness. For a newly proposed task focused on physical realism, the quantitative evaluation is very sparse. The paper does not introduce a comprehensive evaluation benchmark, which is a missed opportunity and makes it difficult to rigorously assess the method's performance across diverse and challenging scenarios.

3.  Based on the few quantitative experiments provided, the method's effectiveness is questionable. The reported improvements over the VACE baseline in standard video quality and identity preservation metrics are marginal. This lack of significant improvement casts doubt on the practical value of the complex proposed framework.

**Questions:**

Please refer to the weakness.

---

### Official Review · Reviewer_bjsB · 2025-10-28

**Soundness:** 2
**Presentation:** 2
**Contribution:** 2
**Rating:** 4
**Confidence:** 4

**Summary:**

This paper proposes ThinkPlace, an end-to-end video editing framework that uses a VLM-guided, chain-of-thought (CoT) “think-then-place” stage to plan physically plausible object-environment interactions before diffusion-based generation. The system couples (i) global semantic reasoning and spatial grounding to produce guidance tokens and edit masks, (ii) a Spatial-DPO post-training scheme that combines local (edit-region) and global (scene-wide) preference optimization via VLM-based ranking, and (iii) an iterative VLM post-evaluation loop for corrective refinement. Experiments against VACE/UNIC indicate improved physical realism (PC/PR), identity preservation, and video quality on diverse scenarios.

**Strengths:**

1. The problem formulation of the lack of physical realism for current video editing methods makes sense.
2. ThinkPlace demonstrates improved physical realism (PC/PR), identity preservation, and video quality in quantitative evaluations.

**Weaknesses:**

1. My biggest concern is that the proposed "thinking" and "chain-of-thought" video editing strategy is overclaimed. The proposed method leverages a frozen MLLM (Qwen2.5-VL) to generate the thinking tokens, which are then sent into a frozen video editing model (VACE) after being transformed by a trainable connector layer. These operations mean that (1) the thinking tokens are generated solely based on the original reasoning capability of the pretrained MLLM, instead of emerging from SFT or RL processes; (2) the video editing model receives only the thinking tokens as a new condition; it does not participate in the thinking or CoT reasoning process. In this regard, this method is more similar to "using an MLLM as a prompt rewriter", instead of doing actual "thinking" or CoT reasoning during video editing.

2. Both the "chain-of-thought" reasoning steps and the DPO preference labels are generated by an MLLM (Qwen2.5-VL). Is there any analysis on the reliability of this MLLM's reasoning capability over physical realism in videos? The MLLM is also responsible for generating the bounding boxes. With only a pretrained 7B MLLM, I believe some of the tasks proposed in the paper will be quite difficult for the MLLM. For example, is there an MLLM-human agreement study to show that the preference labels generated by Qwen2.5-VL are indeed aligned with human preferences? Is there any evidence to show that the MLLM generates the bounding box locations with high accuracy?

3. The descriptions of the data curation pipeline are very vague both in the main text and in the appendix. There is no description of the source for the post-training dataset, and the section lacks citations for the various models used during dataset preprocessing.

**Questions:**

1. "Specifically, the connector learns to project environment-aware Interaction CoT tokens into the T5 embedding space used by
Wan2.1" I didn't fully understand this description. What is the format of the CoT tokens? Are they discrete text tokens (token indices from the Qwen2.5-VL tokenizer), or are they continuous representations from the Qwen2.5-VL decoder (hidden representations before the language modelling head)? Does that mean the connector replaces the T5 encoder, or does the T5 encoder still exist, but with an additional adapter layer mapping these two embedding spaces?

2. What does it mean by "GPT-4o generated captions" when evaluating VACE in the experiment section?

3. What is the prompt used in Qwen2.5-VL to generate the thinking tokens?

---

### Official Review · Reviewer_qDuK · 2025-10-28

**Soundness:** 3
**Presentation:** 4
**Contribution:** 3
**Rating:** 4
**Confidence:** 4

**Summary:**

This paper addresses a key limitation in contemporary video editing: the lack of physical plausibility and environmental awareness when integrating custom subjects into videos. Existing methods often produce visually correct but physically impossible results (e.g., a ceramic mug floating on water). The authors propose **ThinkPlace**, an end-to-end "think-then-place" framework that leverages a Vision-Language Model (VLM) as a reasoning engine to guide a video diffusion model.

**Strengths:**

1. The "think-then-place" paradigm  is a novel and powerful conceptual contribution. It intelligently leverages the physical reasoning capabilities of VLMs to guide diffusion models, effectively bridging the gap between visual quality and physical realism  without requiring explicit physics simulators.



2. The methodology is technically robust and multi-faceted, introducing three key innovations: a VLM-CoT reasoning pipeline for joint semantic and spatial guidance , an automated Spatial-DPO post-training method using VLM-based preference ranking , and an iterative VLM-based corrective refinement loop.

**Weaknesses:**

1. The framework's performance appears highly dependent on the reasoning capabilities of the specific VLM used (Qwen-VL2.5). The paper acknowledges this limitation  but does not include an ablation study on the VLM choice, making it difficult to gauge how performance would scale with less capable or different VLM "brains."

2. The iterative refinement process (Section 4.4) , while effective, adds significant computational latency. The paper states convergence "typically" occurs in 2-3 iterations, but a more detailed analysis of this latency, its distribution across test cases, and potential failure modes (e.g., non-convergence) is missing.

3. The data curation pipeline (Section 3.3) is a complex, multi-stage "reverse-engineering" process involving several models (VLM, DINO-SAM, Bagel). This introduces its own potential sources of error, and the impact of this synthetic-style data on the model's generalization to novel, in-the-wild user requests is not fully explored.

**Questions:**

1. Regarding the VLM dependency: How sensitive is the quality of the Interaction CoT and spatial grounding to the choice of VLM? For instance, what is the performance drop (e.g., in PC/PR scores ) if Qwen-VL2.5  is replaced with a smaller or different VLM?

2. Concerning the iterative refinement (Section 4.4): What percentage of the test videos required 1, 2, or 3+ refinement cycles to converge? Are there failure cases where the VLM post-evaluation gets "stuck" in a diagnostic-refinement loop, and how does the system handle non-convergence?

3. For the Spatial-DPO (Section 3.2) : The consensus-based ranking protocol  is used to reduce evaluation noise. What was the initial agreement rate between the two independent VLM ranking sessions, and how many candidate pairs were discarded due to inconsistent rankings?

4. Could you provide more detail on the connector module (Section 3.1.2)? It projects the Interaction CoT tokens into the T5 embedding space. How effectively does this simple two-layer MLP  preserve the complex, structured physical reasoning from the CoT, and were alternative, more complex projection mechanisms explored?

---

### Official Review · Reviewer_SpD1 · 2025-10-31

**Soundness:** 3
**Presentation:** 3
**Contribution:** 2
**Rating:** 6
**Confidence:** 4

**Summary:**

This paper presents Think Before You Place, a framework that leverages large language models to reason about 3D object placement before predicting locations. By generating semantic reasoning as guidance, It improves the realism, coherence, and interpretability of 3D scene generation compared to prior geometry-based methods.

**Strengths:**

1. The problem formulation is novel and meaningful, introducing reasoning-aware 3D object placement—a rarely explored yet important direction.

2. Effectively integrates LLMs’ semantic reasoning with spatial prediction, bridging language understanding and 3D scene generation.

3. Demonstrates clear improvements in realism and interpretability over conventional geometry- or data-driven baselines.

**Weaknesses:**

1. The visualization and qualitative examples are limited, making it hard to fully assess the model’s generalization and reasoning quality.

2. Some qualitative results show physically implausible behaviors, such as the dragonfly’s wing motion in Fig. 4, which contradicts real-world physics.

3. The evaluation lacks human or perceptual studies, which would better validate the claimed improvements in reasoning quality.

4. The reasoning outputs are not deeply analyzed, leaving uncertainty about how much the generated text truly influences spatial decisions.

**Questions:**

1. How long does inference take for a single case?

2. What is the reason for choosing DPO, and why not use methods such as GRPO?

3. Could the authors show more failure cases to illustrate the model’s limitations better?

---

### Official Review · Reviewer_mqDX · 2025-10-31

**Soundness:** 3
**Presentation:** 3
**Contribution:** 2
**Rating:** 4
**Confidence:** 4

**Summary:**

This paper proposes ThinkPlace, a vision-language-guided video editing framework that achieves physically and visually coherent custom subject integration without explicit physics simulation.
The system introduces three core components (1) a VLM-based Chain-of-Thought (CoT) reasoning module (2) a Spatial Direct Preference Optimization (Spatial-DPO) post-training stage; and (3) an iterative corrective editing loop. Built on VACE/Wan2.1 diffusion transformers and Qwen-VL2.5, ThinkPlace achieves improved physical plausibility and video realism on 200 custom-integration test videos.

**Strengths:**

- The Interaction CoT reasoning explicitly analyzes scene physics, lighting, and motion, generating structured spatial and semantic guidance before synthesis.

- The Spatial-DPO and VLM-based preference evaluation remove the need for human annotation, a practical and scalable design validated by gains in PC/PR (physical realism) metrics.

**Weaknesses:**

- Although ThinkPlace leads on all metrics, several improvements are numerically small and may fall within measurement variance. Therefore, (1) it is not clear how significant is the improvement of the method, and (2) Reporting standard deviations or significance tests would help substantiate the claimed superiority, especially for perceptual and identity metrics.

- In this paper, the experiments focus on short, curated physics-aware scenes. It is unclear how the model behaves under open-domain or high-motion videos, e.g., handheld or complex lighting conditions. Including stress-test examples (hand-held camera, low-light, or multi-object scenes) would demonstrate robustness.

- The entire system’s success depends on VLM inference accuracy—errors in bounding-box localization or physical reasoning could cascade. It should better quantify reasoning reliability, for example by reporting bounding-box IoU against manual annotations or showing failure cases where VLM misguidance causes errors (e.g., incorrect occlusion handling).

- The iterative VLM post-evaluation adds 2–3 refinement cycles per video (Sec. 4.4), which can be computationally expensive.

- While the qualitative figures are provided, user studies or pairwise preference tests could more convincingly validate perceived realism, especially since much of the novelty lies in subjective physical plausibility.

**Questions:**

Please refer to the weakness section.

---

### Note · Authors · 2025-11-14

I have read and agree with the venue's withdrawal policy on behalf of myself and my co-authors.